# Impact of an Acceptance and Commitment Therapy programme on HbA1c, self-management and psychosocial factors in adults with type 1 diabetes and elevated HbA1c levels: a randomised controlled trial

Ingrid Wijk [1], Susanne Amsberg,[2] Unn-Britt Johansson,[1,3] Fredrik Livheim,[4] Eva Toft,[5,6] Therese Anderbro[7,8]

For numbered affiliations see end of article.

**Correspondence to**
Dr Ingrid Wijk;
ingrid.wijk@shh.se

## ABSTRACT

**Objective** To evaluate the impact of an Acceptance and Commitment Therapy (ACT) programme, tailored for people living with type 1 diabetes, on glycated haemoglobin (HbA1c), self-management and psychosocial factors among individuals with HbA1c>60 mmol/mol compared with treatment as usual (TAU).

**Setting** An endocrinologic clinic in Sweden.

**Participants** In this randomised controlled trial, 81 individuals with type 1 diabetes, aged 18–70 years with HbA1c>60 mmol/mol, were randomly assigned to either an ACT group intervention or TAU. Exclusion criteria were: unable to speak Swedish, untreated or severe psychiatric disease, cortisone treatment, untreated thyroid disease and newly started insulin pump therapy. At the 2-year follow-up, HbA1c was measured in 26 individuals.

**Intervention** The ACT programme comprised seven 2-hour sessions held over 14 weeks and focused on acceptance of stressful thoughts and emotions, and to promote value-based committed action.

**Outcomes** The primary outcome was HbA1c, and the secondary outcomes were measures of depression, anxiety, general stress, fear of hypoglycaemia, diabetes distress, self-care activities, psychological flexibility (general and related to diabetes) and quality of life. The primary endpoint was HbA1c 2 years after the intervention programme. Linear mixed models were used to test for an interaction effect between measurement time and group.

**Results** Likelihood ratio test of nested models demonstrated no statistically significant interaction effect ($\chi^2$=0.49, p=0.485) between measurement time and group regarding HbA1c. However, a statistically significant interaction effect (likelihood ratio test $\chi^2$=12.63, p<0.001) was observed with improved scores on The Acceptance and Action Questionnaire in the intervention group after 1 and 2 years.

**Conclusions** No statistically significant difference was found between the groups regarding the primary outcome measure, HbA1c. However, the ACT programme showed a persistent beneficial impact on psychological flexibility in the intervention group. The dropout rate was higher than

### STRENGTHS AND LIMITATIONS OF THIS STUDY

⇒ This study was developed using an interdisciplinary approach, with a randomised control trial design.
⇒ The intervention is theory-based, with roots in behavioural research, and is tailored for living with diabetes.
⇒ This research addresses psychological aspects of living with type 1 diabetes, which has been requested by the diabetes community and in international guidelines.
⇒ The dropout rate in this study affects the possibility to draw conclusions from the findings.

expected, which may indicate a challenge in this type of study.

**Trial registration number** NCT02914496.

## BACKGROUND

Type 1 diabetes is an autoimmune-mediated disease[1] with an increasing global incidence in young people under 20 years of age.[2] Complications of type 1 diabetes involve life-threatening conditions such as hypoglycaemia and ketoacidosis as well as microvascular and macrovascular diseases.[3] National and international guidelines recommend treatment targets for glycaemic level, individually adjusted, to prevent development of complications.[4 5] However, only a minority reach the treatment targets.[6–8]

The pancreatic beta cell destruction in type 1 diabetes brings a life-long dependency on supplying exogenous insulin[3] and along with that a self-management posing high demands.[9] Glucose monitoring, insulin adjustment and managing hypoglycaemia and hyperglycaemia are examples of self-care

activities involved.[10] In addition to the amount of ingested carbohydrates, several factors such as physical activity, hormones, stress and temperature affect the glycaemic level.[10] Consequently, self-management requires knowledge and skills[11] and involves frequent decision-making.[12] Emotional stress related to the management of diabetes, diabetes distress, persists over time[13] and is present among 20%–30% of people with type 1 diabetes.[14] Furthermore, the prevalence of depression is up to threefold higher among individuals living with type 1 diabetes.[15] Moreover, depression in individuals with diabetes is linked with an increased risk of mortality.[16] Several studies have found that psychological health problems and elevated glycated haemoglobin (HbA1c) levels are associated.[17–19] Consequently, there has been a call to address the psychological aspects of living with diabetes.[20 21]

Cognitive–behavioural therapy (CBT) is focused on restructuring maladaptive thoughts to create more functional cognition and behaviours.[22] CBT has shown positive effects on depression in a general population[23] as well as among individuals with diabetes.[24] In a randomised controlled trial (RCT) by Amsberg *et al*, a CBT intervention improved HbA1c and reduced levels of depression and diabetes distress.[25]

Acceptance and Commitment Therapy (ACT) expands on CBT. The approach in ACT suggests that the presence of uncomfortable thoughts and feelings should be accepted and not suppressed or modified owing to the fact that human life is difficult.[26] Despite the existence of uncomfortable thoughts, functional behaviours could pave the way to a life in accordance with a person's values and goals. Being present and aware, and at the same time developing functional behaviours adapted in the current context, is referred to as psychological flexibility.[27] Moreover, psychological flexibility, which ACT aims to increase,[28] is suggested to be a key prerequisite to psychological health.[29] ACT is based on six flexibility processes: acceptance—allowance of difficult thoughts and feelings, defusion—not getting fused with thoughts and feelings, self-as-context—observing thoughts and feelings, attention to the present moment—being mindful of the current internal and external environment, values—stating personal values that may provide direction in life and commitment, committed action—creating behaviours that corresponds to personal values.[27] Previous studies have supported treatment effects of ACT for different mental and somatic health problems.[30] With its focus on acceptance of discomfort, ACT has also been argued to be especially suitable for long-term conditions where distress can be a realistic factor.[31] Furthermore, a recent study found that greater psychological flexibility in adults with type 1 diabetes was associated with less distress and more beneficial HbA1c.[32] A meta-analysis concluded that ACT interventions may improve the ability of self-care, psychological flexibility and HbA1c in type 2 diabetes.[33] Within the scope of type 1 diabetes, few ACT interventions for adults have been evaluated. A feasibility study of a single-arm trial involving individuals with type 1 diabetes and comorbid eating disorder indicated positive effects on psychological flexibility and diabetes distress, in addition to being feasible and improving eating disorder symptoms. Furthermore, an RCT feasibility study of an ACT-based intervention provided in web format included individuals with type 1 and type 2 diabetes. A reduction in diabetes distress was seen in the intervention group. However, the dropout rate was 57% during treatment.[34]

In summary, type 1 diabetes is a severe disease with an increased prevalence of impaired psychological health with treatment targets reached by only a minority. Previous research regarding ACT and diabetes has mainly focused on type 2 diabetes. Our hypothesis was that an ACT programme would lead to improvements in glycaemic and psychosocial outcomes in an adult population with type 1 diabetes and prolonged elevation of HbA1c. Thus, we conducted an RCT aiming to evaluate the impact of an ACT programme, tailored for people living with type 1 diabetes, on HbA1c, self-management and psychosocial factors.

## METHODS
### Study design and participants
This study was a longitudinal two-arm RCT conducted at an endocrinologic clinic in Sweden. The intervention group participated in an ACT programme and the control group received treatment as usual (TAU) only. The endocrinology clinic population comprised adults (>18 years) with type 1 diabetes and included urban and suburban residents. A study protocol has been published where a more detailed description of the method and the intervention is provided.[35]

Eligible participants were identified by extracting data from the Swedish National Diabetes Register. Individuals who met the inclusion criteria: type 1 diabetes with a duration of at least 2 years, aged 18–70 years and HbA1c>60 mmol/mol on two occasions during the preceding year, were assessed in a random order (generated by computer). Sequentially, patient records were reviewed for information on exclusion criteria. The following criteria were applied: unable to speak Swedish, untreated or severe ongoing psychiatric disease, cortisone treatment, untreated thyroid disease, and insulin pump therapy treatment started since <3 months.

### Recruitment
A power analysis before recruitment suggested that at least 56 individuals with type 1 diabetes were needed to reach 80% power for detecting a clinically relevant reduction in HbA1c of 6 mmol/mol (with assumed SD of 9) at a 5% significance level. To allow for dropouts of approximately 24%,[25] the aim was to recruit 70 persons with type 1 diabetes.

From 2016 to 2019, individuals with HbA1c>60 mmol/mol (reference value 42 mmol/mol) were assessed for eligibility (n=622) in five rounds. Study information and an invitation to participate were sent by post to eligible

**Enrolment**

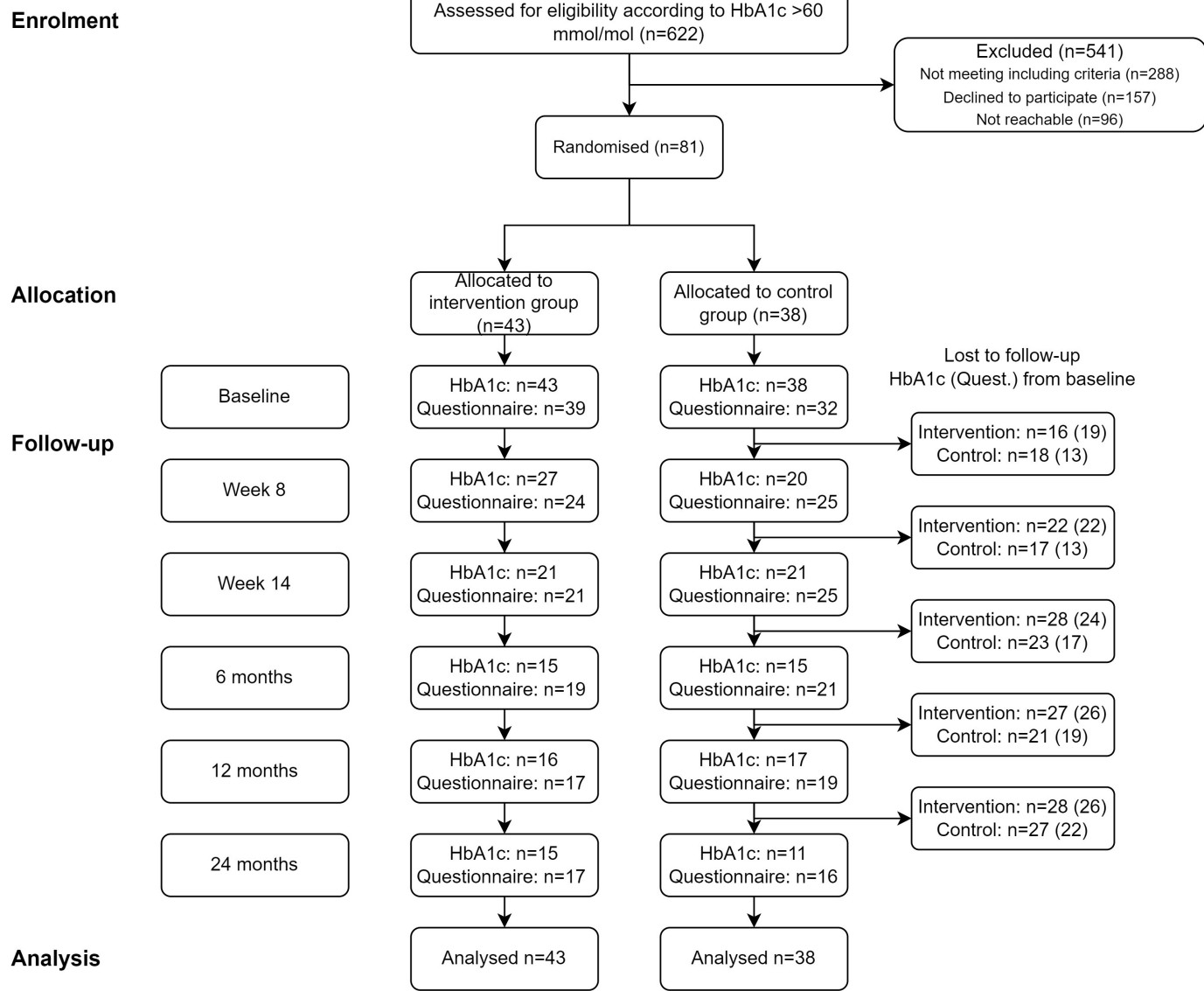

**Figure 1** Flow diagram of recruitment, randomisation and attrition. HbA1c, glycated haemoglobin.

individuals who met the criteria for inclusion (n=343). Two weeks after the material was sent, a research nurse assistant telephoned the invited individuals to offer more information about the study, if desired. Eighty-one people submitted written informed consent and were included in the study. Consequently, the participation rate was 23.6% (81 of 343 who received an invitation). Sociodemographic details and information on treatment were collected at baseline.

### Randomisation
After the baseline measurements, participants were randomly allocated into either the intervention or control group (figure 1). A person not involved with the project conducted randomisation by drawing sealed lots. Stratification was carried out to achieve an equal distribution by sex in the groups. The research assistant nurse informed participants of the group assignment by email and provided further instructions 3 weeks before the start of the intervention programme. Blinding the ACT

instructors or the research assistant nurse to the group allocations during the study period was not possible. However, the ACT instructors never met the control group; additionally, the research nurse assistant was not involved in the intervention programme and did not have ACT skills.

### Intervention
This study was developed by a multidisciplinary team consisting of psychologists, nurses specialised in diabetes and an endocrinologist. The ACT programme consisted of seven 2-hour sessions that were held every other week. A text message was sent the day before each session as a reminder. The size of the ACT groups ranged from six to nine participants. Furthermore, the programme was manual based and developed according to a general ACT programme.[36] The manual was tailored by a psychologist specialised in diabetes and a diabetes nurse to direct the focus to living with type 1 diabetes; both were trained ACT instructors. The sessions involved activities connected to

the six core components of ACT: being present, acceptance, defusion, self as context, values and commitment.[27] Recurring activities throughout the programme were group discussions, psychoeducation, exercises, mindfulness practice, role play and value awareness. There was also homework between sessions.

The intervention group followed the ACT programme alongside TAU and the control group received TAU. TAU comprised a minimum of two visits (one appointment with an endocrinologist and one with a nurse specialised in diabetes) at the clinic each year. The needs of the patient governed the number of visits and telephone contacts.

## Assessments
Time points for measurement were at baseline, weeks 8 and 14 during the intervention and repeated measurements at 6 months, 1 year and 2 years after the intervention. Web-based questionnaires were sent to participants by email and reminders were sent 1 and 2 weeks after the first email, if needed. The questionnaires and reminders were sent via a web-based tool. Three weeks before each follow-up assessment, a letter with information regarding the coming measurements and dates for HbA1c measurement was sent via post and via email. Reminders of the upcoming dates were sent a couple of days before HbA1c measurement, which took place close to the clinic but not in the hospital area. If both the questionnaires and HbA1c measurement were completed, a gift card of 100 Swedish crowns was sent to the participant after each time point of measurement.

### Primary outcome measure
The primary outcome was HbA1c. The measurement reflects the average glycaemic level during the previous 2–3 months.[37] The treatment goal should be individually set; however, a general treatment target for adults is <53 mmol/mol.[38] We used capillary blood samples and equipment to conduct point-of-care immunoassay (DCA Vantage Analyzer, Siemens) which has shown sufficient correlation with the reference method, high-performance liquid chromatography.[39] The method of analysis is calibrated against International Federation of Clinical Chemists standard. The same equipment was used at each measurement time point and a checklist for calibration and maintenance was followed in accordance with the manufacturer's recommendations.[40]

### Secondary outcome measures
#### Depression Anxiety Stress Scales
Depression, anxiety and stress were assessed with The Depression Anxiety Stress Scale (DASS21).[41] The questionnaire consists of 21 items divided into 3 subscales. A 4-point Likert scale, starting with *didn't apply to me at all* and ending with *applied to me very much or most of the time,* is used for rating.[41] Psychometric properties have been evaluated in a Swedish population.[42] Cronbach's alpha in the subscales in the current study ranged from 0.75 to 0.92.

### Hypoglycaemia Fear Survey (HFS)
Fear of hypoglycaemia was assessed by using The HFS.[41 43] The scale with 23 items in total involves 2 subscales: worry and behaviour. A 5-point Likert scale is used for scoring (0=never, 4=always). A higher score indicates a more explicit fear of hypoglycaemia.[43] The Swedish version has been evaluated regarding psychometric properties.[44] Cronbach's alpha for the total scale used in the current study was 0.86.

### Problem Areas in Diabetes Scale (PAID)
The PAID is a measure for assessing emotional distress related to diabetes.[45] The questionnaire consists of 20 items and is scored on a 5-point Likert scale (0=not a problem to 4=serious problem).[45] Greater distress is indicated with higher scores (total score 0–100), where 40 is the cut-off for significant distress.[46 47] The reliability and validity of the Swedish version scale have been evaluated.[48] Cronbach's alpha in the current study was 0.91.

### Summary of Diabetes Self-Care Activities (SDSCA)
The frequency of self-care activities over the past 7 days was measured with the SDSCA.[49] The questionnaire includes 11 items in different areas of self-care activity. The scoring system ranges from 0 to 7 and refers to the number of weekdays. The areas are separately scored and are not summed in a total score. High scores indicate high self-care performance.[49] A Swedish version, translated by Amsberg *et al*, was used which also demonstrated sufficient internal consistency for the subscales (Cronbach's alpha 0.80–0.86) in a similar sample.[25] In the current study, Cronbach's alpha for the subscales diet, exercise and glucose monitoring ranged from 0.80 to 0.90.

### Acceptance and Action Questionnaire (AAQ)
The original AAQ[50] was developed into a revised version, the AAQ-II[51] to measure the degree of psychological flexibility. The revised scale AAQ-II has been translated into Swedish and evaluated regarding psychometric features. This version was used in our study. There are six items in the Swedish version that are rated on a 7-point Likert scale (1=never true, 7=always true). A lower score reflects greater psychological flexibility.[52] Cronbach's alpha in the current study was 0.88.

### Acceptance Action Diabetes Questionnaire (AADQ)
Psychological flexibility related to diabetes was assessed using the AADQ. The questionnaire is derived from the AAQ and was developed by Gregg *et al*.[53] The original AADQ comprises 11 items and is rated on a 7-point Likert scale (1=never true, 7=always true). A higher score indicates a greater psychological flexibility[53] in contrast to the AAQ which is scored in the opposite direction.[52] The scale was translated into Swedish, and the psychometric properties have been evaluated. The manuscript has been accepted, but is not yet published. Cronbach's alpha in the current study was 0.77.

## Manchester Short Assessment of Quality of life (MANSA)

The MANSA is used to assess quality of life (QoL). The questionnaire was developed by Priebe et al[54] and was constructed with 16 items (12 items related to general QoL and 4 are objective items of QoL). The rating score ranges from 1 to 7 (1=could not be worse, 7=could not be better). The response categories in the objective items are dichotomous (yes, no). A Swedish version has been evaluated regarding the psychometric properties.[55] The Cronbach's alpha in the current study was 0.85.

### Statistical methods

Baseline characteristics were described as mean (SD) for continuous variables and absolute and relative frequencies for categorical variables. Tests for differences in baseline characteristics between completers and non-completers and intervention (control group, intervention group) were performed using the Kruskal-Wallis test (continuous variables) and $\chi^2$ test (categorical variables). Mixed effect models were fitted to test for changes in HbA1c levels and questionnaire scores across follow-up. Each model included the time point (baseline, week 8, week 14, week 38, week 62) and intervention (control group, intervention group) as fixed effects and a random intercept for participants. Interaction effects between the time point and intervention were explored. Test of fixed effects were done using a likelihood ratio test in nested models using the $\chi^2$ distribution. We implemented a compound symmetry structure within our mixed effects model. Graphical assessment of residuals plotted against predicted values were conducted to validate the model's assumption.

All statistical analyses were performed using R V.3.6.2.[56] IBM SPSS V.27 was used for graphs (IBM, Armonk, New York, USA). Other statistical analyses that were planned and presented in the study protocol could not be conducted or were not suitable owing to a high number of missing values and lack of statistical significance in outcome. The intention-to-treat principle was applied in the primary analysis. In addition, a per-protocol analysis was conducted in a further examination of the results.

### Fidelity check of the intervention

For quality assurance, a fidelity check was performed by videotaping one session held by the licensed psychologist and the registered nurse. The videotape was reviewed by a psychologist, an experienced trainer who has been training instructors in the ACT group intervention that was used. In addition, participants in two groups and the group leaders estimated the adherence to the treatment intervention and manual by completing an adherence evaluation after sessions three and seven. The aim was to measure how adherent each participant had been to the treatment interventions in the sessions and how well the group leaders adhered to the treatment manual. The group leaders also evaluated their adherence to the manual. Finally, participants' and group leaders' evaluations were compared. Participants and group leaders rated each intervention in the session on a scale from 0 to 10, where 0 = 'this has not been discussed or brought up' and 10 = 'this was covered thoroughly'. The estimates are shown in online supplemental tables 1 and 2.

## PATIENT AND PUBLIC INVOLVEMENT

The RCT was preceded by an informal pretrial among eight volunteers with type 1 diabetes to test the design and content of the ACT intervention. On the basis of the feedback of participants, an adaptation of the programme was made with shorter but more sessions. The participants in the RCT will receive a message when the results are published and where to find them. There will also be a link provided to a webpage with a summary of the results in plain Swedish.

## RESULTS

### Study participants

The mean age for all study participants in the sample was 40.2 (SD 16.6) years and 63% were women. The mean duration of type 1 diabetes was 23.4 (SD 14.4) years, the mean HbA1c was 73.5 (SD 13) mmol/mol and 74.7% of participants have had retinopathy on some occasion. Approximately 88% of participants used sensor-based glucose monitoring and 35.8% wore an insulin pump. In table 1, the baseline characteristics are described for the total sample and the allocated trial groups. No statistically significant differences between group characteristics were found at baseline. Baseline scoring of the secondary outcomes and comparisons between the groups are shown in online supplemental tables 3 and 4. No statistically significant differences were found between the groups in baseline scoring (Bonferroni correction applied). Significantly elevated diabetes distress was present among 51% of the study sample, (PAID Score>40). Mean scores in the normal range were noted for stress and anxiety. The mean score in the depression subscale of 12.14 (SD 10.85) indicated mild depression, and 39% of participants had scores corresponding to moderate-to-severe depression.

### Dropouts and adherence to the programme

Eight participants did not start the intervention programme and wished to withdraw from the study. Reasons for their decision were as follows: internship in another city (n=1), going abroad (n=1), lack of time (n=1), family reasons (n=1) and having to work (n=1). In three cases, no reason was provided. Reasons for interrupting the programme were as follows: disliked the programme (n=2), medical-related/health-related reasons (n=3), evening time not suitable (n=1), family reasons (n=1) and change in working schedule (n=1). One person in the intervention group passed away before the 2-year follow-up. In the control group, the reasons reported for discontinuation of the study after randomisation until the last time point for measurement (2 years) were as follows: medical reasons (n=1), family reasons

**Table 1** Baseline characteristics of trial groups with comparisons between complete (participants completing the 2-year follow-up) and dropout (participants lost to follow-up or who discontinued the study at the 2-year follow-up) groups

| Characteristics | All participants | Intervention group | Control group | P value | Complete | Dropout | P value |
|---|---|---|---|---|---|---|---|
| N | 81 | 43 | 38 | | 26 | 55 | |
| Males, n (%) | 30 (37) | 15 (34.9) | 15 (39.5) | 0.844 | 8 (30.8) | 22 (40) | 0.578 |
| Age (years) (SD, range) | 40.2 (16.6, 19–69) | 42.2 (16.5, 19–69) | 37.9 (16.6, 20–68) | 0.368 | 38.6 (16.4) | 40.9 (16.8) | 0.754 |
| Diabetes duration (years) (SD, range) | 23.4 (14.4, 3–61) | 23.6 (14.0, 3–61) | 23.2 (5, 15–61) | 0.730 | 20.2 (12.5) | 24.9 (15.1) | 0.156 |
| Family/cohabitation | | | | | | | |
| With a partner, n (%) | 36 (44.4) | 17 (39.5) | 19 (50) | | 11 (42.3) | 25 (45.5) | |
| With parents, n (%) | 14 (17.3) | 7 (16.3) | 7 (18.4) | | 3 (11.5) | 11 (20.0) | |
| With other, n (%) | 5 (6.2) | 2 (4.7) | 3 (7.9) | | 0 (0.0) | 5 (9.1) | |
| Alone, n (%) | 22 (27.2) | 13 (30.2) | 9 (23.7) | | 11 (42.3) | 11 (20.0) | |
| Apart, n (%) | 4 (4.9) | 4 (9.3) | 0 (0.0) | 0.314 | 1 (3.8) | 3 (5.5) | 0.167 |
| Education (highest level) | | | | | | | |
| Elementary school, n (%) | 4 (4.9) | 3 (7.0) | 1 (2.6) | | 0 (0.0) | 4 (7.3) | |
| Upper secondary school, n (%) | 30 (37.0) | 14 (32.6) | 16 (42.1) | | 8 (30.8) | 22 (40.0) | |
| University (courses or education) n (%) | 41 (50.6) | 24 (55.8) | 17 (44.7) | | 16 (61.5) | 25 (45.5) | |
| Other, n (%) | 6 (7.4) | 2 (4.7) | 4 (10.5) | 0.441 | 2 (7.7) | 4 (7.3) | 0.361 |
| Employment | | | | | | | |
| Retired, n (%) | 6 (7.4) | 3 (7.0) | 3 (7.9) | | 1 (3.8) | 5 (9.1) | |
| Early retirement, n (%) | 5 (6.2) | 3 (7.0) | 2 (5.3) | | 3 (11.5) | 2 (3.6) | |
| Student, n (%) | 11 (13.6) | 5 (11.6) | 6 (15.8) | | 3 (11.5) | 8 (14.5) | |
| Not currently working for medical reasons, n (%) | 2 (2.5) | 2 (4.7) | 0 (0.0) | | 0 (0.0) | 2 (3.6) | |
| Unemployed, n (%) | 3 (3.7) | 1 (2.3) | 2 (5.3) | | 1 (3.8) | 2 (3.6) | |
| Employed, n (%) | 44 (54.3) | 21 (48.8) | 23 (60.5) | | 15 (57.7) | 29 (52.7) | |
| Manage own company, n (%) | 10 (12.3) | 8 (18.6) | 2 (5.3) | 0.420 | 3 (11.5) | 7 (12.7) | 0.730 |
| CGM or FGM, n (%) | 71 (87.7) | 37 (86) | 34 (89.5) | 0.897 | 24 (92.3) | 47 (85.5) | 0.608 |
| CSII, n (%) | 29 (35.8) | 15 (34.9) | 14 (36.8) | 1.000 | 12 (46.2) | 17 (30.9) | 0.277 |
| BMI (SD, range) | 26.1 (4.7, 17.3–39) | 25.5 (4.5, 17.3–36.5) | 26.8 (4.7, 19.6–39) | 0.250 | 26.6 (3.4) | 25.8 (5.1) | 0.210 |
| Baseline HbA1c (mmol/mol) (SD, range) | 73.5 (13, 60–125) | 71.4 (10.1, 60–118) | 75.9 (15.4, 61–125) | 0.315 | 69.2 (7.2) | 75.5 (14.6) | 0.029 |
| Retinopathy, n (%)* | 59 (74.7) | 30 (73.2) | 29 (76.3) | 0.950 | 17 (65.4) | 42 (79.2) | 0.291 |
| Neuropathy, n (%)* | 23 (29.1) | 13 (31.7) | 10 (26.3) | 0.780 | 9 (34.6) | 14 (26.4) | 0.624 |
| Nephropathy, n (%)* | | | | | | | |
| Microalbuminuria | 4 (5.2) | 2 (5.0) | 2 (5.4) | | 2 (7.7) | 2 (3.9) | |

| Table 1 | Continued | | | | | | |
|---|---|---|---|---|---|---|---|
| Characteristics | All participants | Intervention group | Control group | P value | Complete | Dropout | P value |
| Macroalbuminuria | 2 (2.6) | 1 (2.5) | 1 (2.7) | | 0 (0.0) | 2 (3.9) | |
| No albuminuria | 71 (92.2) | 37 (92.5) | 34 (91.9) | 0.995 | 24 (92.3) | 47 (92.2) | 0.474 |
| Cerebrovascular disease, n (%)* | 1 (1.3) | 1 (2.4) | 0 (0.0) | 1.000 | 0 (0) | 1 (1.9) | 1.000 |
| Coronary artery disease, n (%)* | 3 (3.8) | 1 (2.4) | 2 (5.3) | 0.946 | 1 (3.8) | 2 (3.8) | 1.000 |

*Information missing for some individuals (nephropathy: 4, neuropathy: 2, retinopathy: 2, cerebrovascular disease: 4, coronary artery disease: 2).
BMI, body mass index; CGM, continuous glucose monitoring; CSII, continuous subcutaneous insulin infusion ; FGM, flash glucose monitoring; HbA1c, glycated haemoglobin.

(n=1), no longer interested (n=2), lack of time (n=1) and moved to another city (n=2). Parts of the 1 year and the 2-year follow-up were conducted during the COVID-19 pandemic during 2020–2021. Three participants actively declined the invitation for follow-up measurement because of the pandemic (two who had symptoms and one owing to the risk of infection). However, one HbA1c measurement could be registered from the endocrinology clinic in the same week. Most reasons for discontinuation/loss to follow-up were unknown.

Online supplemental table 5 presents the exposure to ACT. In total, 35 participants attended at least 1 session (81.4%) and 20 participants (46.5%) attended 4–7 sessions of the programme. The attendance for measurement of HbA1c is also shown in online supplemental table 5). In week 8 after session 4, 62.8% of participants in the intervention group and 52.6% of controls completed HbA1c measurement. The dropout rate was approximately equal in the trial arms during the follow-up period. Furthermore, no significant differences in baseline characteristics were detected among participants remaining in the study after 2 years and those who were lost to follow-up (Bonferroni correction applied) (table 1).

## Primary outcome

The mean HbA1c in the intervention group decreased over 2 years from 71.4 mmol/mol to 59.6 mmol/mol. However, the likelihood ratio test of nested models demonstrated no statistically significant interaction effect ($\chi^2$:0.49, p:0.485) between measurement time point and group affiliation regarding HbA1c, as shown in online supplemental table 3. The observed mean HbA1c over time is illustrated in figure 2.

## Secondary outcomes

A significant interaction effect (likelihood ratio test $\chi^2$:12.63, p<0.001) between time and group allocation was found in AAQ-II scores measuring psychological flexibility (online supplemental table 3. This finding indicates a trend of improvement in the ACT group over time. The observed mean AAQ-II Score over 2 years is illustrated in figure 3.

No significant interaction effects between time and group allocation were demonstrated in scores of the PAID, DASS21 (online supplemental table 3), HFS, AADQ, MANSA or SDSCA (online supplemental table 4 and 5).

## Per protocol

Per-protocol analyses were conducted; the results were consistent with findings in the intention-to-treat analyses.

## DISCUSSION

This study evaluated the impact of an ACT programme tailored for individuals living with type 1 diabetes. The observed mean values decreased to a level below 60 mmol/mol in the intervention group during the study period.

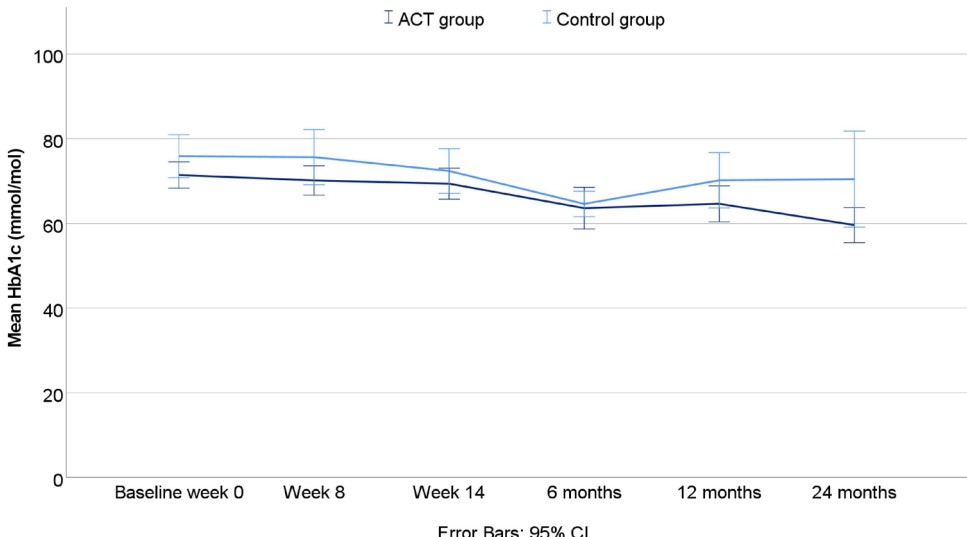

**Figure 2** Mean values of HbA1c over 2 years by group affiliation. ACT, acceptance and commitment therapy; HbA1c, glycated haemoglobin.

However, no statistically significant effect on HbA1c was demonstrated 2 years after the ACT programme. The AAQ-II scores decreased over time and the improvement in psychological flexibility was statistically significant after one and 2 years. The results regarding HbA1c correspond to those of a meta-analysis involving overall psychological interventions in type 1 diabetes[57] and a meta-analysis of CBT interventions for diabetes related distress in type 1 and type 2 diabetes.[58] Some CBT intervention studies solely focusing on type 1 diabetes have shown similar findings.[59 60] At the same time, the meta-analysis of Winkley *et al* and Yang *et al* concluded that CBT has the potential to be effective for HbA1c reduction in type 1 diabetes[57 61] which was clearly demonstrated in the study of Amsberg *et al.*[48] Regarding ACT intervention in diabetes, the findings in our study stand in contrast to those of previous studies. ACT interventions in type 2 diabetes have shown

promising results.[33] A recent study of an ACT intervention for adolescents with type 1 diabetes showed a significant beneficial effect on HbA1c.[62] To our knowledge, this is one of the first RCTs to evaluate the effect of a group-based face-to-face ACT intervention on HbA1c in an adult population. We used a longitudinal design with a long-term follow-up period of 2 years, which can enlighten the mechanism of change over time. The study protocol was developed according to Standard Protocol Items: Recommendations for Interventional Trials 2013 Statement[63] and was previously published and open for transparency and replication. We used instruments that have been evaluated for psychometric properties (with one exception, the translated version of the SDSCA), and the same device for HbA1c measurements.

Interestingly, a persistent beneficial effect was found in psychological flexibility during the follow-up period. The

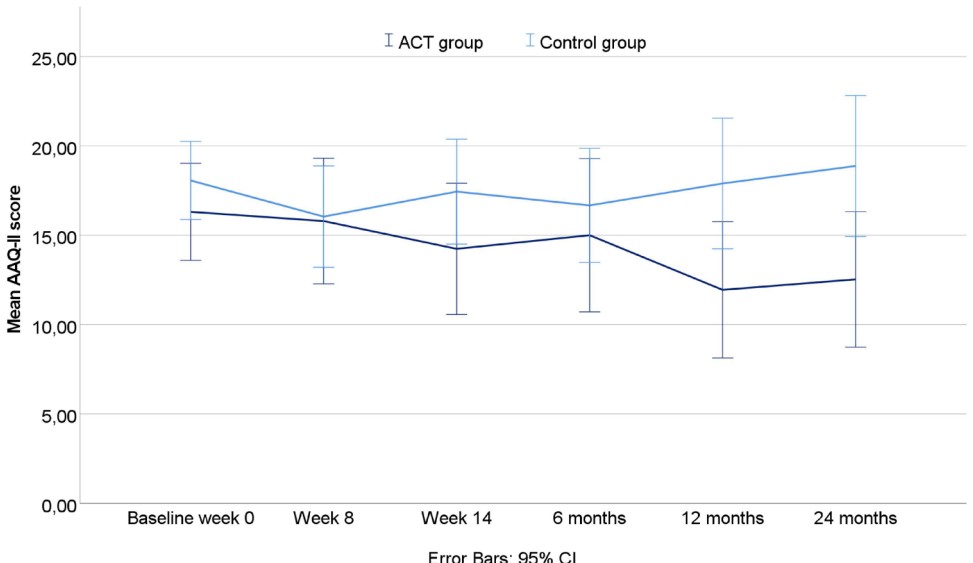

**Figure 3** Mean values of AAQ-II Score over 2 years by group affiliation. AAQ, Acceptance and Action Questionnaire; HbA1c, glycated haemoglobin.

effect was seen 1 year after the intervention but not before that time. This finding may indicate that psychological flexibility takes time to develop; this was also discussed in a previous study.[64] We were unable to conduct mediation analysis regarding the role of psychological flexibility in self-management and HbA1c improvement owing to the absence of statistical significance in the outcome variables. However, a development process that requires time would hypothetically generate improvement in HbA1c even later in the follow-up period. Notably, the statistically significant change in psychological flexibility was only observed in the generic AAQ-II and not in the disease-specific AADQ. The AAQ-II is the most frequently used instrument in research for measuring psychological flexibility, but it has also been criticised for being a measure for psychological distress rather than of psychological flexibility.[65]

Reflections have been made regarding the connection between the content of the psychological intervention and diabetes self-management in explaining outcomes in similar studies.[57 62] Gregg *et al*[53] evaluated the impact of an educational intervention with and without ACT components in type 2 diabetes.[53] The current study did not focus on self-management and did not involve elements of diabetes education, which may have been a disadvantage. Another observed aspect that differentiates the current study from other group-based studies showing HbA1c improvement is the individualised perspective. The intervention in the study by Amsberg *et al*[48] included individual meetings,[25] and Alho *et al*[62] conducted interviews regarding expectations prior to the intervention.[62] Moreover, participants called for more individualisation in a feasibility trial of a web-based ACT programme.[34] An example of a beneficial individualised element in psychological intervention trials is a study by Malins *et al*.[66] A brief motivational interview was introduced after the initial session which increased adherence and decreased the dropout rate in a CBT intervention of chronic pain and cancer.[66]

This study has several limitations, among which the main limitation is the dropout rate. Eight individuals (18.6%) allocated to the ACT group did not start the intervention, which is in line with a meta-analysis of dropout rates in CBT trials.[67] However, the dropout rate during the ACT programme period in the current study far exceeded the aggregated dropout rate in the study by Fernandez *et al*.[67] In the study by Bendig *et al*[34] that involved a web-based ACT programme, the dropout rate was 57% in the intervention group during the study period, which is more similar to our findings. Nevertheless, there are no indications of different dropout rates between trials based on ACT or other psychological interventions.[68] In addition, the COVID-19 pandemic may have affected the attendance for measurement of HbA1c during parts of the 1-year and 2-year follow-up periods. There were no differences in baseline characteristics between participants who remained at the 2-year follow-up and those who were lost to follow-up at the same measurement time point. Without

a Bonferroni correction, the HbA1c demonstrated higher values in the group lost to follow-up which would imply a selection bias. Furthermore, the consequence of the high attrition is an underpowered analysis, which may entail an elevated risk of type II error. Another deficiency in the study is the lack of a comprehensive process evaluation such as the one Craig *et al*[69] described in the Medical Research Council guidelines for complex interventions.[69] On the other hand, we have gained valuable knowledge regarding the context and implementation of ACT groups for adults with type 1 diabetes that lays a foundation for further development. This study indicates that the format of providing tools based on ACT may not be suitable for the targeted study population. The mean age of the participants suggests a life context with demands from family and work life. In addition, previous research has showed that high HbA1c and executive dysfunctions are associated, at least in young adulthood.[70] It is possible that comprehensive programmes for behaviour change which requirements of commitment may be difficult to fit in a busy life. Brief ACT interventions have demonstrated promising results in different disciplines.[71] It is also noteworthy that diabetes distress may be a significant issue for persons with HbA1c within the target range.[72] Thus, interventions targeting diabetes distress, with elements of brief ACT strategies incorporated in diabetes healthcare may be beneficial to explore.

Considering that this was a single-centre study with significant attrition problems, the external validity is weakened. The study is ongoing with the last measurement time point at 5 years after the intervention. The decision to proceed with follow-up measurements despite the high attrition was made owing to the effects of psychological flexibility and tendencies toward a decreasing HbA1c.

In conclusion, this study showed no statistically significant difference between the groups regarding the primary outcome measure, HbA1c. However, the ACT programme showed a persistent beneficial impact on psychological flexibility in the intervention group.

**Author affiliations**
[1]Department of Health Promoting Science, Sophiahemmet University, Stockholm, Sweden
[2]Department of Health Care Sciences, Marie Cederschiöld University, Stockholm, Sweden
[3]Department of Clinical Sciences and Education, Södersjukhuset, Karolinska Institutet, Stockholm, Sweden
[4]Department of Clinical Neuroscience, Karolinska Institutet, Stockholm, Sweden
[5]Department of Medicine, Huddinge, Karolinska Institutet, Stockholm, Sweden
[6]Department of Medicine, Ersta Hospital, Karolinska Institutet, Stockholm, Sweden
[7]Department of Clinical Sciences, Danderyd Hospital, Karolinska Institutet, Stockholm, Sweden
[8]Department of Psychology, Stockholm University, Stockholm, Sweden

**Acknowledgements** The authors would like to thank all participants in the study and RN Neide Johansson for supporting the recruitment and data collection. Furthermore, we would like to acknowledge Fredrik Wiklund (Statisticon AB) for statistical support and psychologist Daniel Ek for the fidelity check of the videotaped material. The authors would also thank Edanz for language editing.

**Contributors** Study conceptualisation and design: TA, SA, FL and ET. Acquisition of data: IW and ET. Analysis and interpretation of data: IW in collaboration with

an external statistician and TA, U-BJ and SA. Manuscript draft: IW. Revision and approval of final draft: TA, U-BJ, SA, ET and FL. Guarantor: TA

**Funding** The study was supported by Lindhés advokatbyrå AB (grant number: LA2016-0429), the Annie and Fritz Tjus donation fund (grant number: N/A), the Sophiahemmet Foundation (grant number: N/A) and the Swedish Diabetes Foundation (grant number: DIA2018-360).

**Competing interests** FL receives royalties from his books written on ACT and income from training professionals in ACT.

**Patient and public involvement** Patients and/or the public were involved in the design, or conduct, or reporting, or dissemination plans of this research. Refer to the Methods section for further details.

**Patient consent for publication** Not applicable.

**Ethics approval** This study involves human participants and was approved by The Swedish Ethical Review Authority (2016/14-31/1). Participants gave informed consent to participate in the study before taking part.

**Provenance and peer review** Not commissioned; externally peer reviewed.

**Data availability statement** Data are available upon reasonable request.

**ORCID iD**

Ingrid Wijk http://orcid.org/0000-0001-8967-1342

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
