## [Reviewer comments · BMJ Open]

ARTICLE DETAILS

TITLE (PROVISIONAL)	Impact of an Acceptance and Commitment Therapy programme on HbA1c, self-management and psychosocial factors in adults with type 1 diabetes and elevated HbA1c levels: a randomised controlled trial
AUTHORS	Wijk, Ingrid; Amsberg, Susanne; Johansson, Unn-Britt; Livheim, Fredrik; Toft, Eva; Anderbro, Therese

VERSION 1 – REVIEW

REVIEWER	Reznik, Yves Centre Hospitalier Universitaire de Caen
REVIEW RETURNED	23-Apr-2023

GENERAL COMMENTS	The paper by Wijk et al aims to evaluate the impact of a cognitive behaviour program on the quality of glucose control, HbA1c being the surrogate, together with other outcomes including psychological flexibility, fear of hypoglycemia, anxiety, depression, diabetes-related emotional distress, self-care ability and quality of life. After completion of the intervention, no favourable effect of the cognitive behaviour program was demonstrated on the primary outcome ie HbA1c, nor on the different psychosocial dimensions which were evaluated, except psychological flexibility which was significantly improved one and two years after completion of the cognitive behaviour program. The aim of the study is interesting and original, since studies on diabetes distress are scarce in the population of people with type 1 diabetes. The design of the trial is rigorous, randomized controlled, including quality control of participant adherence and group leader adherence to the experimental procedure. The Acceptance and Commitment Therapy (ACT) is properly referenced, background studies have shown that similar Therapy was able to positively affect emotional distress and HbA1c level. The large panel of psychological dimensions explored in this rather large cohort is of great interest for understanding the impact of long term living with type 1 diabetes. The major pitfall of the study is the high level of dropout rate which considerably reduced the power of the study analysis. The paper suggests some comments and questions to the authors. 1- Did authors look if correlations are found between the baseline scores exploring diabetes distress, depression, stress, quality of life ?2- The authors may explain more deeply to a non specialist reader the nature of the core components of ACT which were explored.3- Another question concerning the ACT relates to how individual behaviour could be impacted in a group approach. ? Was there any
--

	complementary individual approach by the psychologist or the diabetes nurse ? If no, isn't it a limitation of the ability of the ACT to induce individual behavioral and cognitive changes ? 4- Concerning the Diabetes self-care activities, the scoring system is based on the number of weekdays where the participant was active in different self care domains. How were worded the questions evaluating an active self care attitude concerning diet, glucose monitoring on a precise day ? If I understand the response was Boolean yes/no, does it actually reflect an active self care attitude ? 5- If one exclude the impact of Covid 19 on HbA1c measurement, as acknowledged by the authors, how do they explain the high attrition rate despite several reminders sent to participants ? 6- The level of emotional distress was especially high, since more than 50% participants had a PAID score above 40. Do the authors think that there was a selection bias explaining such level ? Is there a different dropout rate among those with/without severe diabetes distress at baseline ? 7- Similarly did the level of depression, anxiety, stress at baseline influence the high dropout rate ? 8- The authors observed a significant change in the AAQ score suggesting a better psychological flexibility in the experimental group. How do they explain that such change was observed after one year and not between the ACT sessions at weeks 8 and 14 and at 6 months ? one may anticipate a faster effect of the cognitive behavioural therapy...
--	---

REVIEWER	Svensson, Jannet Copenhagen University Hospital, Paediatric Department E None - although I'm working on a project regarding compassion and have some of the same challenges
REVIEW RETURNED	01-May-2023

GENERAL COMMENTS	This is a well-designed and very well conducted study testing the implementation of acceptance and commitment therapy in an adult population of dysregulated patients (HbA1c > 60 mmol/mol) with type 1 diabetes aged 18-70 years. The study design is a randomized controlled trial and participants are invited based on a register subtraction and invitation send after screening for other exclusion criteria such as severe mental illness, steroid treatment etc. The research question is valid and highly relevant. The overall results are highly influenced by the very high amount of drop-outs meaning that 20% in the intervention group never showed up for the first appointment and only 26 of 81 participants completed the study. Initially only 23.6% of those invited ended with an informed consent – meaning there is reluctance in participating in this type of studies. This is in clear contrast to the introduction where all the potential beneficial effects of this type of intervention is described. As the authors write this is in line with other similar studies and both the low participation rate and high number of drop-outs is not a rare situation in this type of study. Given this is a very well-designed study and the invitation is followed by a phone call inviting to ask for more information there is a discrepancy between what health care providers think is the best for the patients and what they are interested in investing time and effort in. They also have sms reminders before each session, so they do try to help individuals remember the appointment. As stated, the design is very nice, the statistics is appropriate and
---

	the presentation is sufficient, meaning I have no problems in recommending going further with publication. But I hope that the authors have the possibility to think of doing something different. There are signs of a positive effect on flexibility and perhaps they will show more benefits in the few they are able to follow for 5 years – but they are a very small % of all our patients with dysregulated diabetes. This means that they will have a highly selected group where it might work – but what about the rest? So why is it so difficult to include and help patients struggling with diabetes with these types of interventions. First, I wonder if HbA1c is the best parameter to select patients. Perhaps screening for diabetes distress and literacy would be more appropriate? Secondly, why do some sign in for the intervention but never show up? Is it possible that they have all the good intentions but their executive function or structure or other things just get more important? Or are they persuaded but not really committed when they sign up? Perhaps there is a conflict in the fact that changing flexibility, commitment and create new good habits takes time and the individuals with high HbA1c would benefit – but they don't have the ability to commit, prioritize diabetes nor stay focused for weeks. So, do we need rethinking of the design or conduction? Do the authors have any possibility to follow-up and get a more in depth understanding of who, why and when do some with dysregulated diabetes commit and others not and who stays in the program? What is needed for more of those struggling to be more flexible, more committed etc. Do we need to work more with acceptance before it is possible to commit? Do we need to have short introduction sessions – only working with some basic things like awareness or being present before any specific intervention or do we need more personalized approach where the first step is to identify each individual's most important struggling point and start there? I hope these thought are useful the paper is well written and well executed – but I just hope to move this area a bit more forward than just another study showing the difficulties in including and keeping participants in this type of intervention that on paper looks very good but in practice is highly challenged.
--	--

REVIEWER	Mathew, Anil P S G Institute of Medical Sciences and Research, Community Medicine
REVIEW RETURNED	19-Jun-2023

GENERAL COMMENTS	 1. My sample size calculation gave a slightly different figure. Also not clear how the non response was initially expected as 24%. 2. In the linear mixed model analysis, which covariance structure was used? 3. Tables and figures should be CONSORT guidelines
---

VERSION 1 – AUTHOR RESPONSE

Reviewer 1

Dear Dr. Yves Reznik,
First of all, thank you for your valuable comments on this paper.

1. We did not examine for possible correlations between baseline variables initially. However, an ongoing project is evaluating correlations and possible predictors regarding baseline variables such as psychological flexibility, diabetes distress and social determinants. We are also aiming for a further investigation of the responsiveness in the study. Consequently, correlations, including your suggestions, will be further explored in a forthcoming study.
2. Thank you for this important comment. We referred to the study protocol where a more profound description of the components can be found. However, we agree that is inconvenient for the reader and that the components should be more clearly described in the paper. We have now added a supplemental description of the ACT components in the background section (please see page 2).
3. Thank you for this relevant question. A meta-analysis concluded support for group based psychological interventions and that the effects were similar to individual therapy on outcomes related to mental health (Rosendahl J, Alldredge CT, Burlingame GM, Strauss B. Recent Developments in Group Psychotherapy Research. *Am J Psychother.* 2021 Jun 1;74(2):52-59. doi: 10.1176/appi.psychotherapy.20200031) . Furthermore, our previous group-based CBT-study demonstrated statistically significant effect on HbA1c (Amsberg S, Anderbro T, Wredling R, Lisspers J, Lins PE, Adamson U, et al. A cognitive behavior therapy-based intervention among poorly controlled adult type 1 diabetes patients--a randomized controlled trial. *Patient Educ Couns.* 2009;77(1):72-80). Qualitative research also points out that peer support is an important component in groups in feeling understood and less alone Due-Christensen M, Zoffmann V, Hommel E, Lau M. Can sharing experiences in groups reduce the burden of living with diabetes, regardless of glycaemic control? *Diabetic medicine : a journal of the British Diabetic Association.* 2012;29(2):251-6. However, elements of individual meetings intertwined in the intervention is a possible source of motivation and engagement in the intervention. We have suggested the lack of individual components as a limitation in the study in the discussion section.
4. In the SDSCA, screening for diabetes self-care activities the questions mainly refer to the preceding seven days. The questions are worded: " On how many of the last SEVEN DAYS did you participate in at least 30 minutes of physical activity? (Total minutes of continuous activity, including walking)." The scale is numeric from 1-7 (days). Nevertheless, we acknowledge that the instrument may be sub-optimal for assessment of diabetes self-care. Considering the advanced technology development in the field of self- management in type 1 diabetes, the accuracy of the SDSCA may be questioned. However, it is one of the few assessment scales there is, developed for this purpose. We will consider you reflections on the choice of instrument in future studies.
5. Thank you for this important question. We have added reflections on potential reasons for dropping out from the study in the discussion section (please see page 15-16).
- 6-7. Considering that the study population had high HbA1c which is associated with high levels diabetes distress, we had expected a high ratio of high PAID scores in the sample. In case of an observational study the high ratio may had entailed a selection bias. The proportion did not statistically differ between the trial groups between baseline which lower the risk of selection bias in the result in this RCT-study. However, we agree that the large attrition may have entailed a selection bias regarding diabetes distress. In the attrition analysis we focused on the primary outcome HbA1c and sociodemographic details. Consequently, we did not look for differences regarding the secondary outcomes between completers and dropouts. In the forthcoming study, we will have a closer look on the secondary outcomes and their role in dropping out from a study. Thank you for this important comment.
8. That is indeed an interesting reflection. Our proposed explanation in the discussion section is that changing psychologic flexibility takes time and therefore become more evident after one year. It is possible that other elements in CBT e.g exposure therapy in a treatment of a certain phobia may initiate a faster process.

Reviewer: 2

Dear Dr. Jannet Svensson,

We are most grateful for your insightful comments! You have pinpointed the challenges of this study and in this research area. We have discussed your reflections and will give them attention in our future project designs. In an attempt to elaborate on these relevant comments, we have added some points for future research (please see page 15-16). We hope that you find this addition to go some way towards the goal of more clinically useful interventions.

Reviewer: 3

Dear Dr Anil Mathew,

Thank you for your observant and important comments.

1. Thank you for this comment. The expected drop out rate was based on our previous study (Amsberg S, Anderbro T, Wredling R, Lisspers J, Lins PE, Adamson U, et al. A cognitive behavior therapy-based intervention among poorly controlled adult type 1 diabetes patients--a randomized controlled trial. Patient Educ Couns. 2009;77(1):72-80). We have now added this information in the manuscript (please see page 3).
2. Thank you for this valuable comment. We used a compound symmetry structure in our model and to validate the model's assumptions, graphical assessment of residuals plotted against predicted values was conducted. For clarity, this was added in the statistical analysis section (please see page 6).
3. We followed the CONSORT checklist while creating tables and figures. However, it was not possible to use the template for the flow chart because of the numerous follow-up occasions. To further conform to the CONSORT flow-chart, we have now added the headlines from the CONSORT template in Figure 1 (please see attached file).

Dear Editor,

We have reviewed the protocol and trial registration to ensure consistent information.

VERSION 2 – REVIEW

REVIEWER	Svensson, Jannet Copenhagen University Hospital, Paediatric Department E
REVIEW RETURNED	06-Oct-2023
GENERAL COMMENTS	I have no further comments - although the abstract could include the notion that higher drop-outs than expected and the challenge in this kind of study.

VERSION 2 – AUTHOR RESPONSE

Dear Dr. Jannet Svensson

Thank you for your comment. We have followed your advice and added a comment in the abstract to address this issue.